# Differential Associations Between Individual Time Poverty and Smoking Behavior by Gender, Marital Status, and Childrearing Status Among Japanese Metropolitan Adults

**DOI:** 10.3390/ijerph22050655

**Published:** 2025-04-22

**Authors:** Mimori Kaki, Hideki Hashimoto

**Affiliations:** Department of Health and Social Behavior, Graduate School of Medicine, The University of Tokyo, Tokyo 113-0033, Japan; hidehashim@m.u-tokyo.ac.jp

**Keywords:** time poverty, smoking behavior, gender gap, parenthood, marital status

## Abstract

Time availability has been recognized as a social determinant of health. However, the association between time poverty and smoking behavior remains to be fully investigated. The aim of this current study was to examine the association between time poverty and smoking behavior by gender, marital status, and childrearing status, which differentially affect time resource availability. We used data from a population-based survey in the greater Tokyo metropolitan area. Participants were men and women aged 25–50 years (N = 2500). Time poverty was defined as a shortage of preferred leisure time compared to actual scheduled leisure time. Descriptive statistics and modified Poisson regression analyses were conducted, and stratified by gender. The study revealed that time poverty may relate to the prevalence of current smoking among single mothers with preschool-age children. However, this trend was not found for men. The findings suggest that time poverty may be heterogeneously associated with smoking propensity depending on gender-bound social roles and resources, which deserves further research for targeting appropriate interventions for health equity.

## 1. Introduction

Smoking is an important risk factor in several types of cancer, cardiovascular disease, respiratory disease, perinatal death, and stillbirth [1]. The smoking prevalence in Japan declined from 20.7% in 2012 to 16.7% in 2019 [2,3], but it remains one of the 10 countries containing two-thirds of the world’s smoking population in 2019 [4]. In addition, there are known social inequalities in smoking rates [5]. Therefore, reducing the smoking population remains an important public health challenge for health equity.

Identifying the living conditions of individuals who smoke is essential for implementing appropriate public health smoking control measures [6]. In addition to socioeconomic resources such as income and education, time is an important finite resource that affects everyday life [7,8]. Lacking the time necessary to maintain a healthy lifestyle is referred to as “time poverty”, and research on the relationships between this concept and other life conditions and health-related activities has been conducted in the broad fields of economics, sociology, and epidemiology [9,10,11]. Although previous studies have examined the relationship between time poverty and smoking, the results are inconclusive [10]. Studies in economics used the resource of time to represent household production function and examined time poverty only at the household level when testing its relationship to smoking. However, time resource allocation is differentially determined across gender and family norms due to gender-role-bound unpaid work duties, and the measurement of time poverty at the household level may not provide a fair assessment of gender differences in time deprivation and subsequent health impacts [12,13]. Theoretically, time resource allocation should be chosen based on an individual’s preference. However, realized time allocation may not necessarily reflect the preference due to external restrictions such as socially obliged duties like childcare. To better address the association between gender-role-bound time constraints and health behavioral choices, we propose in this study to use a time poverty measurement revealing an individual’s preference, of which details will follow shortly, to examine the association between time poverty and smoking as a stress coping behavior. In the next section, we provide a review of the concept and measurement of time poverty, then propose an alternative measure of time poverty, followed by empirical results and discussion.

## 2. Literature Review

### 2.1. Concept and Measurement of Time Poverty

Although the basic concept of time poverty refers to a shortage of time for leisure and self-investment, its operational constructs vary according to the measurement process. There are two main approaches to measuring time poverty: subjective and objective [14].

Studies using subjective approaches often focus on measuring time pressure. Time pressure is the perception of not having enough time to meet one’s obligations [15]. Although many studies have examined this concept, to the best of our knowledge, there is no internationally accepted gold standard measure of time pressure [16], and it is often measured using a single item, such as “Do you often feel rushed and pressed for time?” [9]. An advantage of the subjective approach is that it explores the intensity of the time shortage experienced by individuals. However, subjective measures of time pressure have been reported to be susceptible to recall bias [17]. Furthermore, time pressure measures alone do not include contextual factors such as when and under what circumstances people experience time pressure. Instead, objective measures of time poverty mainly rely on a time diary, in which the respondent is asked to answer actual time use for specific domains of daily activities such as leisure, sleep, work, housework, etc. The domain-specific time diary is known to be robust against recall bias and other types of self-report bias [17].

Once the time diary measurement assessed time resource allocation to specific activities, there are two standards to determine time poverty: absolute and relative [14]. A typical case of the absolute time poverty standard was adopted in an early study by Vickery [18]. In this method, the time spent on basic activities (e.g., sleeping, eating, personal care) and housework is subtracted to obtain the amount of discretionary time. If the discretionary time is less than the time spent on working and commuting, then the individual is considered time poor.

Many previous studies using a relative approach have set the time poverty threshold at 50% to 70% [9,12,14,19,20] of the median of a sample’s leisure time distribution. A previous study by Bó [21] defined the bottom 25% of the time distribution as time poor. The advantage of identifying a relative standard for time poverty is that it can reveal the time required for specific activities straightforwardly [20]. For example, establishing a general standard for the range of sleeping time is easy because everyone needs to sleep for a certain amount of time each day [20]. However, the amount of time needed for childrearing and housework varies depending on the needs of the children, parenting capacity in a household, and the size of the household [14,22,23]. Thus, the obtained threshold may not be similarly applicable to people with various background characteristics, such as social status (employed or unemployed) and household composition [23]. In this study, we propose an alternative measurement of time poverty to reflect an individual’s preference. Theoretically, time allocation given the limited 24 h should be chosen by an individual’s preference to maximize their utility. However, realized time allocation may not achieve the preferred one due to external restrictive conditions such as the socially-obliged duty of childcare, etc. Our measurement is intended to identify time poverty as the gap between the realized time schedule and ideal time resource allocation based on the individual preference of time use.

### 2.2. Time Poverty, Health, and Gender: Evidence and Hypothesis

Time poverty and busyness have been reported to relate to poor mental health [13], lack of sleep [10,13], excessive alcohol consumption [10], and low exercise levels [13,24]. These previous studies assumed that unpaid work is evenly distributed between partners [20]. Previous economic studies treat households as decision-making units that rationally allocate resources such as time and money for household welfare production. However, despite the substantial increase in the number of women in the labor market, gender inequalities remain in the burden of unpaid work, such as housework and childcare [25,26]. Previous time poverty studies on health have not sufficiently considered these gender differences in allocating time to unpaid work.

We suspected that previous studies failed to detect an association between smoking behaviors and time poverty owing to a lack of consideration of the possible heterogeneity in the association across gender and social roles. Time poverty is not experienced uniformly across individuals. Rather, it is shaped by social structures, particularly gender roles related to marital status and parenting responsibilities [12]. Previous study findings suggest that the effect of time poverty on health-related behaviors differs by gender [9,10]. Working women with children often experience a “second shift” of unpaid labor after their paid work hours [12,27], leading to greater constraints on their leisure time compared to their male spouses [28]. As a result, their time constraints may increase the likelihood of engaging in stress-related coping mechanisms such as smoking. Furthermore, single parents who are solely responsible for parenting are more likely to experience time poverty [11], and the smoking rate among single mothers is particularly high [29,30]. Therefore, in this current study, we examined whether the effect of time poverty on smoking behavior differs according to gender and social roles, such as marital status and childrearing status, which would differentially affect time needs and resources.

## 3. Materials and Methods

### 3.1. Data and Analytic Sample

We used data from the first and second waves of the Japanese Study on Stratification, Health, Income, and Neighborhood (J-SHINE). The survey was conducted in 2010 and 2012 and targeted adults aged 25–50 years and their families living in the greater Tokyo metropolitan area. The survey sample was probabilistically selected from the residential registries of each of four municipalities (two in the Tokyo metropolitan area and two in neighboring prefectures). Details of the surveys have been previously published [31]. The baseline survey conducted in 2010 collected basic information on the target population (age, gender, education, and other information), and the 2010 and 2012 surveys collected data on health-related behaviors, including smoking status.

### 3.2. Measures

#### 3.2.1. Explanatory Variable: Time Poverty

The 2012 survey collected information on actual and ideal daily time schedules. We theorized that in an ideal life schedule, the time allocation should match the rational resource allocation, given the individual’s utility function, and that in actual life, time is allocated according to the external constraints on an individual. Thus, the gap between the two should reflect an individual’s non-volitional time poverty [14,20].

First, participants were asked to indicate the average amount of time spent in the last month on activities such as necessary use (meal, personal care, sleep), contracted use (working, commuting), committed activity use (housework, childcare and other informal care, community services, and volunteer activities), physical activity and learning, and leisure time (“hobbies, entertainment, and socializing” and “relaxation other than sleep”) in 10-min increments (Table 1) [23]. If several activities were carried out at the same time, participants were asked to focus their response on the main activity. For the actual time survey, respondents were asked about all activities on weekdays and holidays (days without work). Next, questions were asked about the daily time allocation that the participant thought was ideal, even if it was considered infeasible. The questions on each activity were the same as those for the actual time schedule, except that the ideal time questions only asked about weekday activities.

In both the actual and ideal time surveys, participants were instructed to ensure that the total daily activity time fell between 20 and 24 h. The responses were linearly transformed to make 24 h in total. Then, the time spent on “hobbies, entertainment, and socializing” and “relaxation other than sleep” was summed for both the actual and ideal time schedules, and this total was defined as leisure time. We used the total amount of time available for leisure time in the assessment of time poverty, since we thought the availability of leisure time would be sensitive to smoking as a stress-coping behavior. We calculated time (in)sufficiency using the following formula (Equation (1)):*T_d_* = *T_r_* − *T_i_*(1)

*T_r_*: actual leisure time;*T_i_*: ideal leisure time;*T_d_*: difference between actual and ideal leisure time.

*T_d_* was obtained by subtracting *T_i_* from *T_r_* to determine the gap between actual and ideal leisure times. The threshold for time poverty was set at −1 standard deviation (SD) of the whole *T_d_* distribution. Less than the threshold indicated “with time poverty”.

#### 3.2.2. Outcome Variable

We used participants’ smoking behavior in 2012 as an outcome for the analysis. In the questionnaire, respondents were asked about current and past smoking behavior (i.e., “I still smoke habitually”, “I used to smoke but have stopped now”, and “I have never smoked habitually”). The binary variable for smoking behavior was defined using a value of 0 to indicate a current non-smoker and a value of 1 to indicate a current smoker.

#### 3.2.3. Covariates

As socioeconomic variables, we used age, gender, and educational attainment as potential confounders. Age was treated as a continuous variable, and gender as a binary variable. Educational attainment up to upper secondary specialized training school level was considered as lower educational attainment, and participants who graduated from postsecondary courses in specialized training schools, junior colleges, technical colleges, or university or above were categorized as higher educational attainment.

We assumed that marital status and parenting preschoolers would affect the relationship between time poverty and smoking, because these two factors strongly represent gender-related norms and social roles. We further hypothesized that these three variables would differentially modify the association between time poverty and smoking propensity. Consequently, we conducted analyses stratified by gender and then included marital status and childrearing status as effect modifiers. Marital status was a binary variable, with a value of 0 indicating that the person was married as of 2012 and a value of 1 indicating that the person was unmarried. Participants who had pre-school age children were categorized as 1, whereas those who had no children or had school-age children (elementary school or above) were taken as a reference.

#### 3.2.4. Outliers and Missing Data

The process for treating outliers and missing data is shown in Figure 1. The total number of respondents from the first wave (2010) and second wave (2012) was 2971. The final number of respondents used in the analysis was 2500. Most of the missing data were derived from daily time schedule items. Since the missing patterns were likely to depend on unmeasured variables such as the availability of housework support from spouses and informal/formal sources, we considered them as non-ignorable missing patterns that resist statistical imputation [32]. Consequently, we conducted a complete case analysis.

#### 3.2.5. Analytical Method

Descriptive statistics, including socioeconomic status, smoking status, and time poverty, were obtained for men and women. Smoking rates were estimated by time poverty status for each gender, then further stratified by marital status and having preschool-age children.

Second, we used a modified Poisson regression with a robust variance estimator to examine the association between time poverty and smoking behavior prevalence, stratified by gender. The smoking rate among Japanese women was around 10%, while among men it was around 30% [2]. Given that the odds ratio tends to overestimate the prevalence ratio when the outcome is common, the use of a modified Poisson regression rather than logistic regression is recommended [33]. We first conducted simple regression model analyses that included time poverty, marital status, and having preschool-age children (Model 1). We then ran the model with interaction terms between marital status, having preschool-age children, and time poverty (Model 2), adjusting for age and educational attainment [34]. Descriptive statistical analysis was performed using Stata/SE 17.0 (StataCorp. 2021. College Station, TX, USA) and Microsoft Excel version 16.82 (Microsoft Corporation. 2018. Redmond, WA, USA). Regression analysis was performed using Stata/SE 17.0. The regression result with the three-way interaction terms of time poverty, being unmarried, and having preschool-age children was visualized in a graph for the marginal estimation of the probability of being a smoker, using the “margin” command in Stata/SE 17.0.

## 4. Results

### 4.1. Basic Statistics

Table 2 shows the descriptive statistics by gender. A total of 74.4% of men and 76.7% of women had higher educational attainment. Unmarried status was observed in 29.5% of men and 25.8% of women. A total of 28.3% of men and 26.7% of women had preschool-age children. The percentage of participants who were current smokers in 2012 was 32.6% for men and 10.3% for women. Finally, 16.5% of the population reported experiencing time poverty: 15.9% of men and 16.9% of women.

### 4.2. Smoking Rates by Time Poverty, Stratified by Gender, Marital Status, and Having Preschool-Age Children

Table 3 shows smoking rates by time poverty status stratified by gender, marital status, and having preschool-age children. Marital status (unmarried vs. married: 32.53% vs. 32.66%) and having preschoolers (without preschoolers vs. with preschoolers: 32.22% vs. 33.65%) did not show identical differences in smoking rates among men. Men with time poverty exhibited a higher prevalence than those without (36.31% vs. 31.92%), and married men, especially, experiencing time poverty had a higher smoking rate compared with those without time poverty (38.35% vs. 31.52%). Among women, having preschoolers was associated with a lower smoking rate (no preschoolers vs. having preschoolers: 11.41% vs. 7.36%). Women with time poverty exhibited a 9.01% smoking prevalence compared to 10.60% of their counterparts without time poverty.

### 4.3. Multivariable Regression Results

The results of the modified Poisson regression model analyses are shown in Table 4. Among men, those with a higher education tended to smoke less than those with a lower education (Table 4(1), Models 1 and 2, *p* < 0.01). However, the main effects of time poverty, marital status, and having preschool-age children were not associated with smoking (Model 1). The results did not markedly change after inclusion of interaction terms between time poverty, marital status, and having preschoolers (Model 2).

Women with higher educational attainment tended to smoke less (Table 4(2), Model 1, *p* < 0.01). Additionally, having preschool-age children was associated with a lower likelihood of smoking in Model 1 (*p* = 0.03). In contrast, being unmarried and experiencing time poverty did not show any clear association with smoking (Model 1). These patterns remained consistent in Model 2, which included interaction terms. However, the interaction between being unmarried and having preschool-age children was associated with a higher likelihood of smoking (*p* = 0.07). The three-way interaction that included time poverty, being unmarried, and having preschool-age children showed a moderate positive association with smoking (*p* = 0.12).

### 4.4. Heterogeneous Effects on Smoking Behavior

Figure 2 illustrates the three-way interaction term of time poverty, being unmarried, and having preschool-age children for the probability of smoking in women, based on the results of Table 4(2).

For married women, the probability of smoking was lower among those with preschool-age children, regardless of whether they experienced time poverty. In contrast, unmarried women with preschool-age children had a higher probability of smoking compared with those without preschool-age children, and this trend was particularly pronounced among women experiencing time poverty.

## 5. Discussion

Previous studies have indicated an association between time poverty and several health-related behaviors [9,10,24,35]. However, most previous studies have calculated the size of time deficits on a household basis, which may not adequately assess the association between time poverty and individual health, given the disparity in time availability for different gender-related roles within the household. Therefore, in the current study, we calculated time poverty using estimates of individuals’ ideal and actual leisure time. To the best of our knowledge, this is the first study to examine the heterogeneous relationship between preference-based time poverty and smoking behavior by gender and gender-related roles.

The findings broadly supported our hypothesis that time poverty would be associated with a greater risk of current smoking, and that this association would be stronger for women at risk of time resource shortage caused by single parenting of preschool-age children. The descriptive statistical analysis indicated that having preschool-age children alone had a negative effect on the smoking rate for women, and the results of the modified Poisson regression confirmed this pattern. However, this association was not found for men. This result is consistent with a previous study reporting that the presence of dependent children had a protective effect on women’s smoking, whereas no clear trend was seen for men [36].

Being unmarried alone did not have any significant association with smoking for either men or women. However, the two-way interaction term combining being unmarried and having preschool-age children indicated a significantly higher prevalence of smoking for women compared to their male counterparts. In addition, when time poverty was combined with being unmarried and having preschool-age children, the slope of this three-way interaction was steeper than that of the two-way interaction between being unmarried and having preschool-age children.

Although we measured time poverty using an objective scale, this measure may reflect participants’ perceptions of non-volitional time excesses and deficiencies (pressure), as it assessed the deviation from participants’ ideal leisure time. A previous study reported that mothers are more likely to experience time pressure than fathers when caring for a new baby, and that such time pressure is related to a deterioration in mental health [37]. Raising a preschool-age child typically increases the burden on mothers and makes it difficult for them to spend time alone [38]. Furthermore, low self-efficacy and a lack of social support are associated with stress in single mothers [39]. Our findings may suggest that for single women with little social support, the protective effect of having preschool-age children against smoking is weakened, and the overwhelming psychosocial disadvantage of time poverty may promote their smoking behavior.

The strength of our study may be seen in two points. First, this study proposed an alternative method for assessing time poverty in the context of individual preference and his/her external constraints due to social roles. Second, we conducted high-order interaction analysis to reveal heterogeneity in the association between time poverty and smoking propensity. We believe these two points contribute to the study of time as a social determinant of health in the context of gender inequality in health and behaviors.

Our findings have several implications for smoking control, especially for parents who have few economic, social, and time resources. To relieve time constraints, it may be useful to improve systems to provide childcare support and psychological support for vulnerable parents. For example, improving affordable access to services and support for caregivers, such as childcare subsidies, expansion of childcare facilities, and universal preschool education, may be worth policy consideration [40]. It may also help to develop supportive social relationships and opportunities for social connections [41]. Furthermore, healthcare professionals should recognize that parents who experience shortages in time and socioeconomic resources may be more likely to smoke. Obtaining information about an individual’s social and economic background, as well as their medical history, may help to provide optimal care tailored to their social context [42]. These implications require further research into time poverty and its heterogeneous impact across genders and gender-bound unpaid workload with a larger and wider population for effective policy translation to reduce smoking risks among vulnerable subpopulations.

## 6. Limitations

The current study involved several limitations related to data, study population, and methodology. Causal inference between participants’ time poverty status and smoking behavior was not plausible due to the cross-sectional design. Because the target population of J-SHINE is limited to the Tokyo metropolitan area and its neighboring prefectures, the results of the current study may not be directly applicable to other areas.

The time diary survey is known to be robust against recall bias and social desirability bias despite its self-reporting nature [14]. However, the remaining recall bias could not be completely denied. Even in surveys with 24-h response times, data from participants with greater time constraints owing to multiple social roles and responsibilities tend to be underestimated [43].

The three-way interaction terms between marital and childbearing statuses and time poverty among women did not reach the conventional level of statistical significance in our results, which may need careful interpretation. While the test of three-way interaction is useful for investigating the heterogeneity effect, it often suffers from low statistical power for small sample sizes and low measurement reliability [44]. In such a case, detection of slope difference, as we presented in Figure 2, may help identify theoretically interpretable associations. We believe the heterogeneous impact of time poverty by gender-related roles is worthy of further investigation with a larger sample. There is evidence that smoking questionnaires for pregnant women tend to underestimate smoking rates in Japanese women in the perinatal period [45]. Thus, it is possible that the negative association between having preschool-age children and current smoking behavior observed in this study was overestimated.

Finally, we were not able to include psychological and social support variables as mediators because we focused on examining the relationship between time poverty and smoking behavior. These also require further investigation.

## 7. Conclusions

The results of the present study indicated that the association between smoking and time poverty is heterogeneous across genders and gender-related social roles. Further investigation targeting the subpopulation with the risk of time poverty and other socioeconomic resource deficiencies would be worthwhile for identifying the leverage to effectively help behavioral change.

## Figures and Tables

**Figure 1 ijerph-22-00655-f001:**
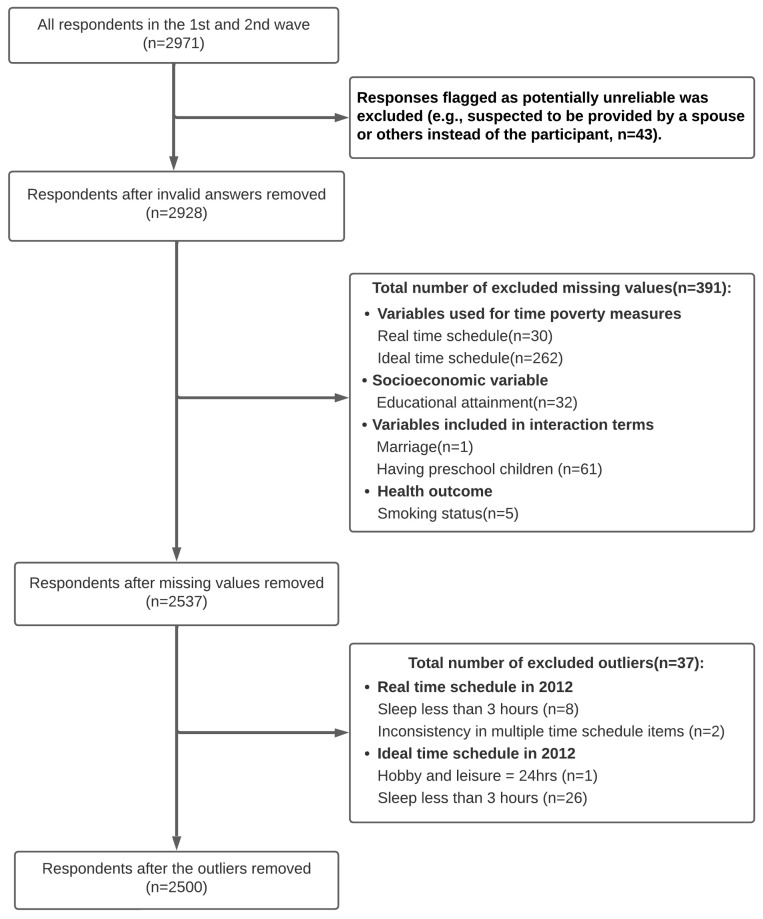
Flowchart of participant selection. A total of 2971 individuals were initially included. After applying exclusion criteria, 2500 participants were included in the final analysis.

**Figure 2 ijerph-22-00655-f002:**
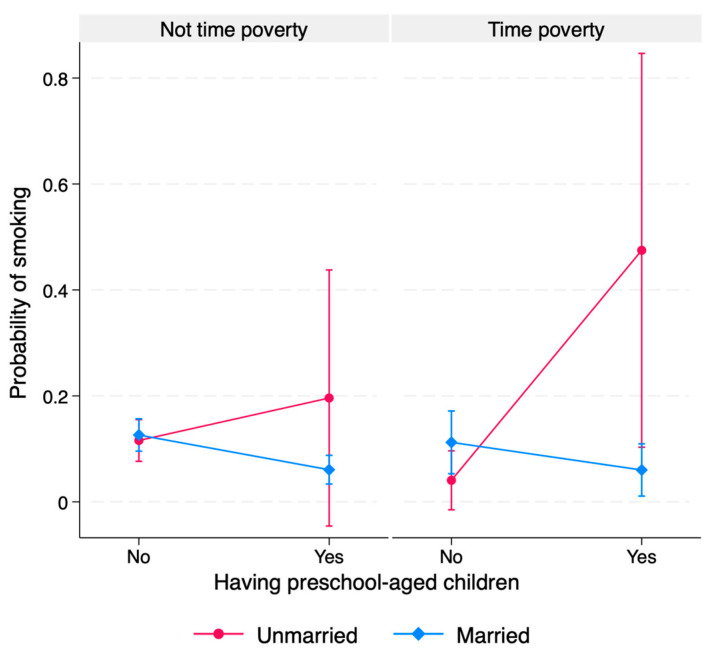
Adjusted predictions for the three-way interaction of marital status, having preschool-age children, and time poverty status among women.

**Table 1 ijerph-22-00655-t001:** Time-use diary questionnaire used in the J-SHINE.

	Your Daily Time Use on Weekdays					
Commuting to work or school	X	X	hrs.	X	0	min.
Work	X	X	hrs.	X	0	min.
Housework and shopping for daily necessities	X	X	hrs.	X	0	min.
Childcare	X	X	hrs.	X	0	min.
Caregiving (parents, spouse, family)	X	X	hrs.	X	0	min.
Exercise, sports, walking	X	X	hrs.	X	0	min.
Learning and lessons	X	X	hrs.	X	0	min.
Community service, volunteer, political activities	X	X	hrs.	X	0	min.
Hobbies, entertainment, socializing	X	X	hrs.	X	0	min.
Rest and relaxation (excluding sleep)	X	X	hrs.	X	0	min.
Sleep duration	X	X	hrs.	X	0	min.
Meals, personal care, etc.	X	X	hrs.	X	0	min.
Total time per day	X	X	hrs.	X	0	min.

Note: The same format was applied to actual and ideal time use diaries.

**Table 2 ijerph-22-00655-t002:** Basic sample characteristics.

	Total (N = 2500)		Men (N = 1125)		Women (N = 1375)	
	N	%	N	%	N	%
Age (mean ± SD *)	37.6	7.0	37.9	7.0	37.3	7.0
Educational level						
High	1892	75.7	837	74.4	1055	76.7
Unmarried	687	27.5	332	29.5	355	25.8
Having preschool-age children	685	27.4	318	28.3	367	26.7
Current smoker	509	20.4	367	32.6	142	10.3
Time poverty	412	16.5	179	15.9	233	16.9

Note: A low level of educational attainment, such as high school graduation or lower. * SD, standard deviation.

**Table 3 ijerph-22-00655-t003:** Smoking rates by time poverty status, gender, marital status, and having preschool-age children.

	Men					Women				
	Marriage		Having Preschool-Age Children		Total	Marriage		Having Preschool-Age Children		Total
	No	Yes	No	Yes		No	Yes	No	Yes	
Without time poverty	32.87%	31.52%	31.45%	33.09%	31.92%	11.73%	10.18%	11.73%	7.12%	10.60%
With time poverty	30.43%	38.35%	36.03%	37.21%	36.31%	8.33%	9.19%	9.52%	8.14%	9.01%
Total	32.53%	32.66%	32.22%	33.65%	32.62%	11.27%	10.00%	11.41%	7.36%	10.33%

**Table 4 ijerph-22-00655-t004:** Modified Poisson regression of smoking risk with interaction of time poverty, unmarried status, and having preschool-age children: **(1)** men. **(2)** women.

	Model 1				Model 2 (Including Interaction Terms)			
	Coef.	*p*-Value	95% CI		Coef.	*p*-Value	95% CI	
			Lower	Upper			Lower	Upper
**(1)**								
Age	−0.00	0.80	−0.02	0.01	−0.00	0.75	−0.02	0.01
Educational level								
Junior high school/high school (ref.)								
College/graduate school	−0.51	0.00 ***	−0.68	−0.34	−0.51	0.00 ***	−0.68	−0.35
Married								
Yes (ref.)								
No	0.04	0.70	−0.18	0.27	0.08	0.52	−0.16	0.33
Has preschool-age children								
No (ref.)								
Yes	0.05	0.62	−0.16	0.27	0.09	0.48	−0.15	0.33
Time poverty								
No (ref.)								
Yes	0.08	0.48	−0.14	0.29	0.20	0.17	−0.09	0.50
Interaction terms								
1. Unmarried × Preschool-age children					0.26	0.65	−0.86	1.37
2. Time poverty × Unmarried					−0.28	0.33	−0.83	0.27
3. Time poverty × Preschool-age children					−0.23	0.38	−0.74	0.28
4. Time poverty × Unmarried × Preschool-age children					NA	NA	NA	NA
**(2)**								
Age	−0.02	0.11	−0.05	0.01	−0.02	0.08	−0.05	0.00
Educational level								
Junior high school/high school (ref.)								
College/graduate school	−1.03	0.00 ***	−1.34	−0.72	−1.01	0.00 ***	−1.32	−0.70
Married								
Yes (ref.)								
No	−0.03	0.89	−0.46	0.40	−0.09	0.71	−0.53	0.36
Has preschool-age children								
No (ref.)								
Yes	−0.53	0.03 **	−0.99	−0.07	−0.73	0.01 **	−1.26	−0.21
Time poverty								
No (ref.)								
Yes	−0.18	0.42	−0.61	0.25	−0.21	0.68	−0.67	0.44
Interaction terms								
5. Unmarried × Preschool-age children					1.26	0.07 *	−0.12	2.64
6. Time poverty × Unmarried					−0.93	0.23	−2.45	0.59
7. Time poverty × Preschool-age children					0.11	0.85	−0.98	1.20
8. Time poverty × Unmarried × Preschool-age children					1.83	0.12	−0.47	4.12

* *p* < 0.1; ** *p* < 0.05; *** *p* < 0.01. CI, confidence interval. Age was a continuous variable; all other variables were binary. The reference groups for each interaction term, corresponding to the numbering above, are as follows: (1) married and no preschool-age children, (2) no time poverty and married, (3) no time poverty and no preschool-age children, and (4) no time poverty, married, and no preschool-age children.

## Data Availability

The original data presented in the study are openly available via FigShare at [https://doi.org/10.6084/m9.figshare.27927471.v1] (accessed on 18 April 2025).

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
