# Peer review of "Differential Associations Between Individual Time Poverty and Smoking Behavior by Gender, Marital Status, and Childrearing Status Among Japanese Metropolitan Adults"

_ijerph, 2025, doi:10.3390/ijerph22050655_

Round 1
Reviewer 1 Report
Comments and Suggestions for Authors
This manuscript examines the relationship between individual-levelly measured time poverty (difference between ideal and actual leisure) and participation in smoking for men and women in Tokyo. It also analyses these variations by marital status and childrearing status. The manuscript is interesting, in that it uses ideal and objective leisure time to measure time poverty, and links it to smoking behaviour, and examines variations by important key life events; marital status and fertility. I believe this manuscript can be improved significantly by providing greater detail and justification to the reader about the case of Tokyo, situating itself in the literature of time poverty, and the expected hypotheses of variations by gender, marital status and fertility.

Author Response
Comment 1-1; When the authors say that they propose a measure of time poverty at the ‘individual-level’ (e.g., line 48), I think it is a bit confusing/misleading. I personally believe the authors mean that they contribute to a new way of measuring time poverty, either by incorporating both ideal/subjective and objective measures of leisure, or they contribute by using ideal leisure time as a new individual level threshold to measure time poverty.
Even if time poverty was previously measured using the median household leisure time of the sample of interest (or a % threshold) as the baseline, the measurement of time poverty itself can be at the individual level e.g., my time poverty measure is X and my friend's is Y, but we use the same threshold baseline measure. In some time use research for example, time poverty can be measured by dividing time diary data for each individual into ‘necessary’, ‘contracted’, ‘committed’, and leisure time (see Wiliams, Masuda and Tallis, 2016 in Social Indicators Research). Of course, the debate of how to measure time poverty is still ongoing and this manuscript has the ability to contribute to it, but the terminology needs to be understood immediately. Perhaps the authors are trying to say that they use individual time poverty measures to examine the association between time poverty and smoking — if this is the case, this is not clear.
Response; Thank you for your thoughtful comment and helpful reference information. We realized that our use of the phrase “individual-level time poverty measure” may be misleading. We admit that time schedules are measured at the individual level by using time diary in previous studies as well as in our own. Instead, thanks to the reviewer’s comment, we recognized that the uniqueness of our measurement proposal is sought in our focus on individual’s revealed preference; existing measures rely on the leisure time amount with a fixed threshold (e.g., median of leisure time) to define time poverty. Theoretically, time allocation given the limited 24 hours should be chosen by individual’s preference to maximize their utility. However, realized time allocation may not achieve the preferred one due to external restrictive conditions such as the socially obliged duty of childcare, etc. Our measurement intended to identify the time poverty as the gap between realized time schedule and ideal time resource allocation assumingly based on the individual preference of time use. We have revised the manuscript accordingly to improve clarity and accuracy in line 42 to 56 as follows. We also changed the term “individual time poverty” to “time poverty” or “preference-based time poverty” instead. Finally, we limited to the lack of leisure time for assessing time poverty, rather than to take a broader “discretionary” time use including “committed”, since we thought a lack of leisure time would be more sensitive to stress the coping behavior of smoking. We would like to refer to Williams, et al. 2016 in this regard.
Original (Introduction; L 41-50)
Studies in economics have used the resource of time to represent household production function, and have examined time poverty only at the household level when testing its relationship with smoking. We speculate that time resource allocation is socially determined across gender and family norms, and that measurement of time poverty at the household level may not provide an accurate assessment of gender differences in time deprivation. Instead, we propose that it is appropriate to use individual-level time poverty to examine the association between time poverty and smoking behavior. In the next section, we provide a review of the concept and measurement of time poverty. We then propose an alternative measure of time poverty at the individual level, followed by empirical results and discussion.
- Revised;
Studies in economics have used the resource of time to represent household production function, and have examined time poverty only at the household level when testing its relationship with smoking. However, time resource allocation is differentially determined across gender and family norms due to gender-role bound unpaid work duties, and that measurement of time poverty at the household level may not provide a fair assessment of gender differences in time deprivation, and subsequent health impact [12,13]. Theoretically, time resource allocation should be chosen based on an individual’s preference. However, realized time allocation may not necessarily reflect the preference due to external restrictions such as socially obliged duties like childcare. To better address the association between gender-role bound time constraint and health behavioral choice, we propose in this study to use a time poverty measurement revealing individual’s preference, of which details will follow shortly, to examine the association between time poverty and smoking as a stress coping behavior. In the next section, we provide a review of the concept and measurement of time poverty, then propose an alternative measure of time poverty, followed by empirical results and discussion. 
Original (Section 3.2.1 Explanatory variable; time poverty; L146-150)
First, participants were asked to indicate the average amount of time spent in the last month on activities such as working, commuting, housework, childcare and other informal care, physical activity, sleep, and leisure time in 10-minute increments. The total time available for “hobbies, entertainment, and socializing” and “relaxation other than sleep” was treated as the amount of leisure time respondents had at their disposal.
- Revised
First, participants were asked to indicate the average amount of time spent in the last month on activities such as necessary use (meal, personal care, sleep), contracted use (working, commuting), committed activity use (housework, childcare and other informal care, community services and volunteer activities), physical activity and learning, and leisure time (“hobbies, entertainment, and socializing” and “relaxation other than sleep”) in 10-minute increments [23].
We use the total time amount available for leisure time in the assessment of time poverty, since we thought availability of leisure time would be sensitive to smoking as a stress coping behavior.
Citation #23; Williams J.R., Masuda Y.J, Tallis H. A Measure Whose Time has Come: Formalizing Time Poverty." Social Indicators Research 2016:28(1);265-283. DOI: 10.1007/s11205-015-1029-z
Comment 1-2; The Literature Review needs substantive improvement on these major points: There is critique on line 68 about how subjective measures using time diaries have recall bias, but is unclear how the measure and the data they use alleviates this issue. Recall bias is again presented in the authors' limitations.
Response; Thanks again for your clarification. We should have been clearer when we mention which measurement method is more susceptible to recall bias. In the sentence of line 68, when we wrote as “the unavoidable risk of recall bias is a disadvantage of this method,” what we meant by “this method” was a subjectively measured time pressure, not time-use diary. We found the sentence was misleading, and would like to correct properly, with apologies for confusion. We meant by “subjective” a measure relying on individual’s perception, following Kalenkoski, et al. (reference 14), which is known to be susceptible to reporting biases. Time pressure is often measured using a single-item question that captures participants’ subjective sense of being pressed for time, depending on their role duties and related norm pressure, which may be prone to reporting bias. In contrast, the present study uses data obtained from a time-use diary, which Kalenkoski, et al. (ref14) categorized as “objective” measurement. Although it relies on self-report rather than objectively observed time use, this method involves domain specific use of time in daily schedule, which is reportedly less vulnerable to recall bias (Kalenkoski, et al. ibid). In addition, in our proposed measurement where the respondent was asked to answer ideal as well as real time schedules, it gives the respondents to carefully reflect on their daily time schedule which is expected to help a precise recall. However, we could not completely deny the possibility of remaining recall bias (reference 43 te Braak, et al. 2023). We added in method section and discussion section on how the time schedule assessment using domain-specific diary could reduce the chance of recall bias while mentioning a possibility of remaining bias.
Original (Section 2.1 Concept and measurement of time poverty; L66-69)
Subjective measures of time pressure often involve time-use diaries [8, 14]. However, the unavoidable risk of recall bias is a disadvantage of this method [15]. Furthermore, time pressure measures alone do not include contextual factors such as when and under what circumstances people experience time pressure [16].
- Revised;
Subjective measures of time pressure have been reported to be susceptible to recall bias [17]. Furthermore, time pressure measures alone do not include contextual factors such as when and under what circumstances people experience time pressure. Instead, objective measures of time poverty mainly rely on time diary, in which the respondent is asked to answer actual time use for specific domains of day activities such as leisure, sleep, work, housework, etc... The domain specific time diary is known to be robust against recall bias and other types of self-report bias [17].
Original (Section 2.1 Concept and measurement of time poverty; L71-72)
One frequently used method for developing objective time poverty criteria follows an early study by Vickery [17].
- Revised
Once time diary measurement assessed the time resource allocation by specific activities, there are two standards to determine time poverty: absolute and relative [14]. A typical case of absolute time poverty standard was adopted in an early study by Vickery [18].
Original (Section 2.1 Concept and measurement of time poverty; L97-103)
However, it may be inappropriate to use the same criteria for people who differ in background characteristics, such as social status (employed or unemployed) and household composition [23].
- Revised
However, the amount of time needed for childrearing and housework would vary depending on the needs of the children, parenting capacity in a household and the size of the household [14, 22, 23]. Thus, the obtained threshold may not be similarly applicable to people with various background characteristics, such as social status (employed or unemployed) and household composition [23]. In this study, we propose an alternative measurement of time poverty to reflect an individual’s preference. Theoretically, time allocation given the limited 24 hours should be chosen by individual’s preference to maximize their utility. However, realized time allocation may not achieve the preferred one due to external restrictive conditions such as the socially obliged duty of childcare, etc. Our measurement intended to identify the time poverty as the gap between realized time schedule and ideal time resource allocation based on the individual preference of time use.
Original (Section 6. Limitations; L377-401))
Our study data were obtained using a computer-aided personal instrument that allows respondents to enter their answers at their convenience. Even in surveys with 24-hour response times, data from participants with greater time constraints owing to multiple social roles and responsibilities tend to be underestimated [42].
- Revised;
Our study data were obtained using a computer-aided personal instrument that allows respondents to enter their answers at their convenience. The time use data derived from the time diary survey is known to be robust against recall bias and social desirability bias despite its self-report nature [14]. However, the remaining recall bias could not be completely denied. Even in surveys with 24-hour response times, data from participants with greater time constraints owing to multiple social roles and responsibilities tend to be underestimated [43]
Comment 1-3; It is also a bit odd that only economics studies are mentioned, what about other disciplines like epidemiology and public health? Are there genuinely no other disciplines that study time poverty/restriction and smoking?
Response; Since the concept originates from economics, the majority of references are from economic studies. However, the reviewer correctly mentioned time poverty has also been studied in fields such as sociology and epidemiology as we also cited studies on time poverty and life style behaviors (sleep, smoking, etc.) in sociology (e.g., #21 Bó B, 2022), and epidemiology (e.g., #9 Strazdins et al, 2016). We have modified the sentence in line 39 to 40 of the manuscript to reflect this broader scope.
Original (Introduction; L37-39)
Lacking the time necessary to maintain a healthy lifestyle is referred to as “time poverty,” and research on the relationships between this concept and other life conditions and health-related activities has been conducted mainly in the field of economics [8-10].
- Revised
Lacking the time necessary to maintain a healthy lifestyle is referred to as “time poverty,” and research on the relationships between this concept and other life conditions and health-related activities has been conducted in the broad fields of economics, sociology, and epidemiology [9-11].
Comment 1-4. Section 2.2 needs to be expanded to discuss why there may be these differences in the relationship between smoking and time poverty by gender, marital status, and childbearing status. Examining these heterogeneities need to be justified clearly. Otherwise, I could easily throw in other potential intersections like education and time poverty. This was discussed briefly in lines 194 onwards but needs to be discussed upfront, and the reason for these hypothesized differences need to be situated in the literature.
Response; We appreciate this comment. We realized we should have clarified the rationale for our focus on time poverty and gender-related time schedule restriction with smoking propensity. Time poverty is deeply shaped by social structures, particularly gender roles, marital status, and parenting responsibilities (#12 Chatzitheochari and Arber, Br. J. Sociol. 2012 for the argument), which directly influence the allocation of unpaid labor. In the current study we specifically focused on the fact that social roles assigned to individuals based on gender have an even greater impact on the distribution of daily time. Marital and childbearing status significantly impact daily time distribution and the burden of unpaid work, which would cause gender difference in time poverty impact on their lives. If time poverty is related to smoking as a stress coping strategy, we hypothesized that it would differentially impact on women with obliged roles of unpaid work, which makes our rationale to include interaction terms between time poverty and marital and childbearing statuses stratified by gender. Since we did not make the rationale above explicit in the Introduction part, we added our focus on gender-related time constraints and subsequent stress management to the study objective as we responded to your comment 1-1.
As the reviewer commented, while education is strongly related to smoking prevalence, and it may influence time management skills, we believe education is to be treated as a confounder to address the association between time poverty and smoking, rather than an effect-modifying factor of time poverty on smoking. We also added the argument above in the Section 2.2 as below.
Newly added (Introduction; L43-50)
- Please refer to our response to Comment 1-1.
Original (Section 2.2 Time poverty, health, and gender; evidence and hypothesis L115-125)
In addition, we suspected that previous studies failed to detect an association between smoking behaviors and time poverty owing to a lack of understanding of the heterogeneity of this association across gender and social roles. Previous study findings suggest that the effect of time poverty on health-related behaviors differs by gender [8, 9], and it is important to consider gender differences when examining time poverty [8, 28]. Furthermore, single parents who are solely responsible for social parenting are more likely to experience time poverty [10], and the smoking rate among single mothers is particularly high [29, 30].
- Revised
In addition, we suspected that previous studies failed to detect an association between smoking behaviors and time poverty owing to a lack of understanding of the heterogeneity of this association across gender and social roles. Time poverty is not experienced uniformly across individuals; rather, it is shaped by social structures, particularly gender roles related to marital status and parenting responsibilities [12]. Previous study findings suggest that the effect of time poverty on health-related behaviors differs by gender [9, 10]. Working women with children often experience a "second shift" of unpaid labor after their paid work hours [12, 28], leading to greater constraints on their leisure time compared to their male spouses [27]. As a result, their time constraints may increase the likelihood of engaging in stress-related coping mechanisms such as smoking. Furthermore, single parents who are solely responsible for social parenting are more likely to experience time poverty [11], and the smoking rate among single mothers is particularly high [29, 30].
Comment 1-5. Why is Tokyo a case study of interest? The study cites smoking data on Japan in 2019 but the data is in 2012
Response; This study conducted a secondary data analysis using the data derived from the JSHINE survey, a population-based survey targeting individuals living in the greater Tokyo metropolitan area for the purpose to address social determinants of health across socioeconomic and demographic strata (please see Takada et al., J Epidemiol, 2014). The reason we used this dataset was simply that the second wave of JSHINE in 2012 included time diary data with information on sociodemographic and socioeconomic conditions that we needed for the current study purpose, and this kind of dataset was not available otherwise as we know of in Japan. We do acknowledge the limited generalizability, this point has also been noted in the limitations section of the manuscript.
As the reviewer suggested, we should have presented the general smoking rate trend to match the time of the data obtained. The smoking rate in 2012, as reported by the National Health and Nutrition Survey of Japan, has been added to lines 28–29 in the main text with due references.
Comment 1-6. In Section 3 materials and methods, better justification about why a modified Poisson regression is used. An example of what the time diary questionnaire looks like would be helpful to understand how ideal leisure was measured (in 10-minute intervals?) perhaps in the Supplementary Material. 
Response; Following the advice, we have added a note regarding the reason for adopting modified Poisson regression by referring to Cummings 2009.
We also newly present the format used for time use diary in Table 1. The format was commonly applied to both actual and ideal daily schedules. This comment also reminds us that we failed to describe what makes “leisure time” in our dairy categories, and we added the description that leisure time was the sum of the time categorized into “hobbies, entertainment, and socializing” and “relaxation other than sleep.”
Original (Section 3.2.5 Analytical method L222-230)
Second, we used modified Poisson regression to examine the association between time poverty and smoking behavior risk stratified by gender.
- Revised
Second, we used modified Poisson regression with a robust variance estimator to examine the association between time poverty and smoking behavior risk, stratified by gender. The smoking rate among Japanese women is around 10%, while among men around 30% [4]. Given that the odds ratio tends to overestimate the prevalence ratio when the outcome is common, the use of modified Poisson regression rather than logistic regression was recommended [33].
Comment 1-7. Under Section 4 Results, the three-way interaction term is not statistically significant, the p-value is 0.12.
Response; Thanks for noting. We acknowledge that the three-way interaction did not reach a conventional level of statistical significance. However, as Dawson and Richter pointed out, we believe this result may still provide a meaningful insight (please refer to Dawson & Richter, 2006, Journal of Applied Psychology). In this article, Dawson and Richter argued that detecting three-way interactions with high statistical power requires a large sample size and high reliability of measurements which are often not feasible. Instead, the authors recommended to visualize the slope difference to interpret if the difference is theoretically interpretable.
In our study, given the limited sample size and the inherent difficulty of detecting higher-order interaction effects, we interpret the observed slope difference across marital and childbearing statuses for time poverty, as visualized in Figure 2, which would indicate a potential interaction worthing further investigation. However, we do acknowledge that the lack of conventional statistical significance requires cautious interpretation. We added the points in the study limitation section as follows. We also modified the expression of conclusive remarks accordingly;
- Newly added (6. Discussion, study limitation)
The three-way interaction terms between marital and childbearing statuses and time poverty among women did not reach the conventional level of statistical significance in our results, which may need a careful interpretation. While the test of three-way interaction is useful for investigating effect heterogeneity, it often suffers low statistical power for small sample size and low measurement reliability [cite Dawsan and Richter 2006]. In such a case, detection of slope difference as we presented in Figure 2 may help identify theoretically interpretable associations. We believe the heterogenous impact of time poverty by gender-related roles would be worthy of further investigation with a larger sample.
Original (Section 7. Conclusion)
The present study identified a moderate positive association between time poverty and smoking behavior among single mothers with preschool-age children. The results suggest that background factors linked to women’s smoking behavior are related to their disadvantaged psychosocial status as single mothers of preschool-age children who have little free time for themselves. Future smoking research should investigate the availability of resources in this population, particularly leisure time at the individual level.
- Revised
The results of the present study indicated that the association between smoking and time poverty is heterogenous across genders and gender-related social roles. Further investigation targeting the subpopulation with the risk of time poverty and other socioeconomic resource deficiency would be worthy for identifying the leverage to effectively help behavioral change.
Newly cited;
Dawson, J. F.; Richter, A. W. Probing three-way interactions in moderated multiple regression: Development and application of a slope difference test. J Appl. Psychol. 2006,91(4), 917–926. https://doi.org/10.1037/0021-9010.91.4.917
Comment 1-8. Under Section 4.5 Heterogeneous effects on smoking behaviour, confidence intervals need to be added into Figure 2, so we know whether these terms are significantly different from each other.
Response; Following the advice, we added the 95% confidence intervals in Figure 2.
Comment 1-9. Another limitation is the justification of the measure of subjective leisure time — is this the best way to capture it? What are the pros and cons of this measure? While it is interesting in that it is new, there should be more discussion dedicated into justifying its measure.
As we argued in the response to your comment 1-2, we meant by “subjective” a measure relying on individual’s perception, following Kalenkoski, et al. (reference 14), which is known to be susceptible to reporting biases. Time pressure is often measured using a single-item question that captures participants’ subjective sense of being pressed for time. The time pressure question is prone to reporting bias because it is affected by the respondent’s social role duties and related norm pressure. This is the very reason we did NOT choose to use this subjective measure in the study. By contrast, the present study relied on data obtained from a time-use diary, which Kalenkoski, et al. categorized as “objective” measurement. Although it relies on self-report rather than objectively observed time use, this method involves domain specific use of time in daily schedule, which is reportedly less vulnerable to recall bias, and most widely used for the purpose of time use studies (Kalenkoski, et al. ibid). In addition, in our proposed measurement where the respondent was asked to answer ideal as well as real time schedules, it gives the respondents to carefully reflect on their daily time schedule which is expected to help a precise recall. However, we could not completely deny the possibility of remaining recall bias. We added in method section and discussion section on how the time schedule assessment using domain-specific diary could reduce the chance of recall bias. We have noted the potentially remaining bias in the study limitation section. Please refer to our response to comment 1-2.
Minor comments
- Decimal places should be standardised in all tables (see tables 4-1 and 4-2)
- Decimal places have been standardized to two decimal digits in Tables 4.
- If marital status is a binary variable =1 if they are married, why are the interaction terms ‘unmarried’?
- We revised in that unmarried coded as 1. Thanks.
- The reference groups for the interaction terms should be included.
- We added a table legend to indicate the reference groups for interaction terms in Tables 4-1 and 4-2.
- Time poverty (-) and (+) in Table 2, 3-1 and 3-2 are not clearly labelled — why not just write ‘insufficient leisure’ and ‘sufficient leisure’ or just something clearer.
- As we describe in the original L187-189, we took the gap between ideal and real leisure time, then dichotomized the value to the time poverty dummy at -1 standard deviation of the value distribution. The description may not match the notation we used in Tables, which may confuse the reviewer. We added the description on how “time poverty dummy” was labeled in Section 3.2.1 to match the labels in Tables accordingly.
- Why are p-values reported in Table 2 but not in Table 3-1 and 3-2?
- We presented Tables 2 and 3-1,2 for descriptive purposes, while we relied on significance tests by modified Poisson regression model as depicted in Table 4. Thus, we realized that Table 2 may be redundant and removed in the revision.
- Merge Tables 3-1 and 3-2 and separate them with a line between Males and Females so that there is one comprehensive Table on Smoking rates for both male and female respondents by time poverty status, marital status, and having preschool-age children.
- Following your advice, we merged Tables 3-1 and 3-2, and the resulting table has been renamed as Table 3.
Reviewer 2 Report
Comments and Suggestions for Authors
Peer Review Comments
Reviewer’s Assessment of this study
The study "Differential Associations Between Individual Time Poverty and Smoking Behaviour by Gender, Marital Status, and Childrearing Status among Japanese Metropolitan Adults" offers a timely and intriguing look into the intricate relationship between smoking habit and time poverty. The study's focus on various demographic dimensions, such as gender, marital status, and childrearing status, provides a more nuanced view of how these factors may affect smoking habits in a particular urban context. The manuscript perfectly sets the scene by comprehensively explaining smoking behaviour and time poverty. It emphasises how significant these problems are to public health and how important it is to look at their relationships from a demographic perspective. The thorough literature evaluation identifies the gaps this study seeks to fill by consulting current and pertinent research. The study has a solid foundation thanks to the well-stated research aims, which also fit nicely with the title.
The study uses a sound approach that is appropriate for the research issues. Given the binary nature of the outcome variable and the requirement for precise relative risk estimates, modified Poisson regression is suitable for examining the relationships between smoking behaviour and individual time poverty. The results are guaranteed to apply to Japanese urban adults since the study population and sample size selection criteria are well-justified. The researchers conduct the statistical analysis methodically, making suitable modifications for potential confounding factors, and carefully describe the data-gathering procedures. The well-structured results section presents the main findings understandably and rationally. Tables and figures improve the results 'ability, making it simple to comprehend how the connections vary among various demographic groupings. The study suitably presents the statistical significance of the results and supports the conclusions with p-values and confidence ranges.
The writers carefully analyse the results and place them in the context of the body of existing literature in the discussion section. They successfully draw attention to how their findings may affect public health policy and intervention tactics, especially when it comes to addressing smoking behaviour and time poverty across various demographic groups. Along with acknowledging the study's shortcomings—such as possible biases and confounding variables—the discussion makes recommendations for future research directions. The manuscript is well-written and cohesive, with a straightforward narrative guiding the reader through the study's techniques, procedures, findings, and implications. The writing is flawless, clear, succinct, error-free, and professional in tone. The study adds to the body of evidence already available on smoking behaviour and time poverty, providing pertinent and helpful information for legislators and public health professionals.
In conclusion, this study is an important and adeptly attempt to investigate the various relationships between smoking behaviour and time poverty among Japanese adults living in cities. Because of the careful attention to detail in the study design, data collection, and analysis, as well as the careful interpretation of the results, this work will be a significant contribution to the field of public health research.
However, here are my few comments:
1. Merge Tables 3-1 and 3-2 and separate them with a line between Males and Females so that there is one comprehensive Table on Smoking rates for both male and female respondents by time poverty status, marital status, and having preschool-age children.
2. What are the implications of this study's findings? How can the study about the varying relationships between smoking behaviour and individual time poverty guide the creation of focused public health initiatives meant to lower smoking rates among particular populations? By answering this query, the study may be able to give more detailed recommendations to public health professionals and politicians on how to create treatments that are suited to the particular requirements of different demographic groups, thus increasing the research findings' usefulness.
3. What are this study's strengths? To grasp the study strengths further, consider the following query: How does the thorough analysis of the various relationships between smoking behaviour and time poverty, along with the study methodology, support the validity and dependability of the results? Resolving this issue would enable the study to explicitly state its methodological advantages, boosting trust in the reliability and validity of the findings.
4. From your study findings, what recommendations/public health interventions do you think are appropriate for this study population and can be replicated in other studies and settings?
5. Remove the future studies from the conclusion section and make it a separate section with the sub-theme: Future studies for directions. Explain in detail the directions of this study's findings for future studies.
6. The entire manuscript should be sent to a Professional English Editor for proofreading and edits.
7. Overall, it is a perfect paper with well-defined methods and re-defined and modified statistical methods.
Comments on the Quality of English Language
The entire manuscript should be sent to a Professional English Editor for proofreading and edits.
Author Response
Comment 2-1. Merge Tables 3-1 and 3-2 and separate them with a line between Males and Females so that there is one comprehensive Table on Smoking rates for both male and female respondents by time poverty status, marital status, and having preschool-age children.
Response; We had a similar comment from another reviewer. Following the advice, we merged Tables 3-1 and 3-2, and the resulting table has been renamed as Table 3.
Comment 2-2. What are the implications of this study's findings? How can the study about the varying relationships between smoking behaviour and individual time poverty guide the creation of focused public health initiatives meant to lower smoking rates among particular populations? By answering this query, the study may be able to give more detailed recommendations to public health professionals and politicians on how to create treatments that are suited to the particular requirements of different demographic groups, thus increasing the research findings' usefulness.
Response; Thanks for the encouraging comment. Our study suggests that public health efforts for smoking control should focus on the subpopulation who may lack sufficient time resources with limited social and economic resources. Our results identifying single mothers with child-bearing duties having a larger impact of time poverty on smoking behavior may suggest that supporting this subpopulation through improving access to affordable childcare and parental support may help reduce parental stress and time constraints, potentially lead to lower smoking prevalence. It also implies that healthcare professionals should consider social, economic, and time resources of the clients when providing smoking cessation support to ensure tailored interventions. However, these implications require further research on time poverty and its heterogenous impact across genders and gender-bound unpaid workload. We added further focused investigation is needed to better identify effective leverage for helping smoking reduction in vulnerable populations with fewer socioeconomic and time resources.
Original (Abstract)
Time poverty was defined as a gap between actual and ideal daily leisure time schedule. Descriptive statistics and modified Poisson regression analysis were conducted, stratified by gender. The study revealed that individual time poverty may increase the risk of current smoking among single mothers with preschool-age children. However, this trend was not found for men. The findings suggest that time poverty is a risk factor for smoking in women with poor time and social resources. Further research will be necessary to investigate the mechanisms by which time poverty affects smoking behavior to inform the implementation of appropriate interventions.
- Revised
Time poverty was defined as a shortage of preferred leisure time compared to actual leisure time schedule. Descriptive statistics and modified Poisson regression analysis were conducted, stratified by gender. The study revealed that time poverty may relate to the prevalence of current smoking among single mothers with preschool-age children. However, this trend was not found for men. The findings suggest that time poverty may be heterogeneously associated with smoking propensity depending on gender-bound social roles, which deserves further research for targeting appropriate interventions for health equity.
Original (Section 5. Discussion; L380-L392)
Our findings have several implications for smoking control, especially for parents who have few economic, social, and time resources. There is a need to improve systems to provide childcare support and psychological support for parents. For example, services and support for caregivers, such as childcare subsidies, expansion of childcare facilities, and universal preschool education, should be strengthened [39]. Regarding psychological support for parents, it is important to alleviate feelings of loneliness and develop supportive social relationships, and to provide opportunities to create social connections in places that parents visit, such as public libraries and schools [40]. Furthermore, healthcare professionals should recognize that parents who experience shortages in time and socioeconomic resources may be more likely to smoke. Obtaining information about an individual’s social and economic background, as well as their medical history, may help to provide optimal care tailored to their social context, including referrals to non-medical institutions such as local organizations [41].
- Revised
Our findings have several implications for smoking control, especially for parents who have few economic, social, and time resources. To relieve time constraint, it may be useful to improve systems to provide childcare support and psychological support for parents. For example, improving affordable access to services and support for caregivers, such as childcare subsidies, expansion of childcare facilities, and universal preschool education, may be worth policy consideration [40]. It may also help to develop supportive social relationships and opportunities for social connections [41]. Furthermore, healthcare professionals should recognize that parents who experience shortages in time and socioeconomic resources may be more likely to smoke. Obtaining information about an individual’s social and economic background, as well as their medical history, may help to provide optimal care tailored to their social context, including referrals to non-medical institutions such as local organizations [42]. These implications require further research on time poverty and its heterogenous impact across genders and gender-bound unpaid workload with a larger and wider population for effective policy translation to reduce smoking risks among vulnerable subpopulations.
Comment 2-3. What are this study's strengths? To grasp the study strengths further, consider the following query: How does the thorough analysis of the various relationships between smoking behaviour and time poverty, along with the study methodology, support the validity and dependability of the results? Resolving this issue would enable the study to explicitly state its methodological advantages, boosting trust in the reliability and validity of the findings.
Response; Thanks for the advice and recognition of strength in our study. We believe the strength of our study is found two folds. Firstly, this study proposed an alternative method of assessing time poverty in the context of individual preference and his/her external constraints due to social roles. Secondly, we conducted high-order interaction analysis to reveal heterogeneity in the association between time poverty and smoking propensity. We believe these two points would contribute to the study of time as a social determinant of health in the context of gender inequality of health and behaviors. We would add the point in the discussion section as study strength.
Newly added (Section 6. Discussion)
The strength of our study may be sought in two points. Firstly, this study proposed an alternative method of assessing time poverty in the context of individual preference and his/her external constraints due to social roles. Secondly, we conducted high-order interaction analysis to reveal heterogeneity in the association between time poverty and smoking propensity. We believe these two points would contribute to the study of time as a social determinant of health in the context of gender inequality of health and behaviors.
Comment 2-4. From your study findings, what recommendations/public health interventions do you think are appropriate for this study population and can be replicated in other studies and settings?
Response; Thanks again for your encouragement. Please refer to our response to your previous comment 2-2.
Comment 2-5. Remove the future studies from the conclusion section and make it a separate section with the sub-theme: Future studies for directions. Explain in detail the directions of this study's findings for future studies.
Response; We removed the mention about future study from the conclusion. For future direction. Please also refer to our response to your comment 2-2.
Comment 2-6. The entire manuscript should be sent to a Professional English Editor for proofreading and edits.
Response; Although we had a professional English editor check at the first submission, we tried again at the revision submission.
Reviewer 3 Report
Comments and Suggestions for Authors
The manuscript seems to be meaningful. I have some comments.
- Overall:Because an association between smoking status and time poverty in same year was investigated, it is better to write “prevalence” rather than “risk” for smoking status. In other words, it is possible that smoking status can affect time poverty, it is better not to use the word “risk factor” in this analysis.
- 2.1:It is better to use subscripts for Td, Tr, and Ti.
- 2.4:What does “non-ignorable missing patterns resisting statistical imputation” mean? If missing patterns are non-ignorable, a missing data analysis method such as imputation is needed.
- Table1: It is better to write number and proportion for each category of categorical variables such as educational level, marital status, and so on.
- Table 4: It is better to show prevalence ratio (i.e. exp(coef)) instead of coef. Same for 95%CI.
In addition, I think that # is not generally used for describing an interaction.
- Table 4: It might be better to conduct a subgroup analysis rather than an analysis with interaction. If the authors want to emphasize an effect of time poverty among single women with preschool children, it is better to restrict the target population to those in the subgroup analysis.
- Figure 2: Could you explain how the probability of smoking in Figure 2 was obtained in Methods? Were they obtained from the estimates of intercept and coefficient?
Author Response
Comment 3-1; Overall: Because an association between smoking status and time poverty in same year was investigated, it is better to write “prevalence” rather than “risk” for smoking status. In other words, it is possible that smoking status can affect time poverty, it is better not to use the word “risk factor” in this analysis.
Response; Following the advice, the terms “risk factor” and “risk ratio” in the main text have been replaced with “associated factor” and “prevalence ratio,” respectively.
Comment 3-2; Section 3.2.1: It is better to use subscripts for Td, Tr, and Ti.
Response; We have revised the scripts as subscripts.
Comment 3-3; Section 3.2.4: What does “non-ignorable missing patterns resisting statistical imputation” mean? If missing patterns are non-ignorable, a missing data analysis method such as imputation is needed.
Response; Thank you for your comment. The phrase “non-ignorable missing patterns resisting statistical imputation” refers to situations in which the probability of missingness depends on the unobserved (i.e., missing) values themselves—a condition known as Missing Not At Random (MNAR) (Schafer & Graham, Psychol. Methods, 2002).
Under MNAR, the missingness mechanism is non-ignorable, meaning that standard statistical imputation methods assuming Missing at Random (MAR)—such as multiple imputation (MI) or maximum likelihood (ML)—may produce biased results unless the missingness process itself is modeled directly. This is because the missing values in the time-use diary are likely to be influenced by unmeasured confounding factors, such as the division of household responsibilities with a spouse or gender norm attitudes, which was not directly measured in our dataset. Therefore, we judged that applying imputation to this dataset would not be justified. We added a phrase mentioning non-ignorable missing patterns corresponds to MNAR, and not suitable for imputation solely relying on measured variables.
Original (Section 3.2.4; Line 209-211)
Most of the missing data were derived from daily time schedule items, and were considered non-ignorable missing patterns resisting statistical imputation.
- Revised
Most of the missing data were derived from daily time schedule items. Since the missing patterns were likely to depend on unmeasured variables such as the availability of housework support from spouses and informal/formal sources, we considered it as non-ignorable missing patterns that resists statistical imputation [Schafer and Graham, 2002]. Consequently, we conducted complete case analysis.
- #32 Schafer, J.L; Graham, J. Missing data: our view of the state of the art. Psychol Methods 2002, 7(2), 147-77.
Comment 3-4; Table1: It is better to write number and proportion for each category of categorical variables such as educational level, marital status, and so on.
Response; Thank you for your suggestion. In the Table, all the categorical variables are binary, and given the total number presented in the top upper row, we believe the information the reviewer suggested to present is readable from the current form of the table.
Comment 3-5; Table 4: It is better to show prevalence ratio (i.e., exp(coef)) instead of coef. Same for 95%CI.
Response; We appreciate the comment. Indeed, using modified Poisson regression with a robust variance estimator is for obtaining prevalence ratio. However, we chose to use coefficient rather than exp(coef) because we wanted to show the heterogenous association between smoking propensity and time poverty status across gender-bound role statuses (e.g., marital and child-bearing statuses). For this purpose, we thought the presentation of estimated coefficients rather than translated numbers of prevalence ratio is more informative because “prevalence ratio” of interaction terms may not be ready for direct interpretation.
Comment 3-6; Table 4: It might be better to conduct a subgroup analysis rather than an analysis with interaction. If the authors want to emphasize an effect of time poverty among single women with preschool children, it is better to restrict the target population to those in the subgroup analysis.
Response; Thank you for your suggestion. In this study, we used interaction terms to examine whether the impact of time poverty on smoking differs between single mothers with preschool children and other groups. Limiting the analysis to a specific subgroup—such as single women with preschool-aged children—would not allow us to assess whether the effect of time poverty varies by individual social background. One of the key objectives of this study is to investigate how the effect of time poverty on smoking behavior differs according to gender, marital status, and parental status. Therefore, we considered it appropriate to use an interaction-term approach for the analysis. Thanks for your understanding.
Comment 3-7; Figure 2: Could you explain how the probability of smoking in Figure 2 was obtained in Methods? Were they obtained from the estimates of intercept and coefficient?
Response; Thank you for your question. The probabilities in Figure 2 were derived from the estimated results of Poisson regression presented in Table 5-2. More specifically, we used the postestimation command of STATA “margin.” This allows estimation of the probability of smoking at the margin of unmarried status (0-1), childbearing status (0-1), and time poverty status (0-1), adjusting for age and educational attainment. We have added the following sentence to lines 326–328 in the manuscript:
Newly added (Section 3.2.5 Analytic method)
The regression result with the three-way interaction terms of time poverty, being unmarried, and having preschool-age children was visualized in a graph for the marginal estimation of probability of being a smoker using “margin” command in STATA.
Round 2
Reviewer 3 Report
Comments and Suggestions for Authors
Thank you for the revision.